Parallelisation of equation-based simulation programs on heterogeneous computing systems

http://orcid.org/0000-0002-9972-7546 Nikolić Dragan D. dnikolic@daetools.com
DAE Tools Project , Belgrade , Serbia
Petzold Linda
Electronic publication date: 2018 Aug 13
Publication date: 2018
Volume: 4
Electronic Location ID: e160
Received 2018 Mar 22; Accepted 2018 Jul 20
Copyright: © 2018 Nikolic
Copyright year: 2018
Copyright holder: Nikolic
License: This is an open access article distributed under the terms of the Creative Commons Attribution License, which permits unrestricted use, distribution, reproduction and adaptation in any medium and for any purpose provided that it is properly attributed. For attribution, the original author(s), title, publication source (PeerJ Computer Science) and either DOI or URL of the article must be cited.
License URL: https://creativecommons.org/licenses/by/4.0/

Keywords: Modelling, Simulation, Heterogeneous computing, Parallel computing, Differential-algebraic equations, Equation-based, Streaming processors, OpenCL, OpenMP

Funding: The authors received no funding for this work.

==============================
Numerical solutions of equation-based simulations require computationally intensive tasks such as evaluation of model equations, linear algebra operations and solution of systems of linear equations. The focus in this work is on parallel evaluation of model equations on shared memory systems such as general purpose processors (multi-core CPUs and manycore devices), streaming processors (Graphics Processing Units and Field Programmable Gate Arrays) and heterogeneous systems. The current approaches for evaluation of model equations are reviewed and their capabilities and shortcomings analysed. Since stream computing differs from traditional computing in that the system processes a sequential stream of elements, equations must be transformed into a data structure suitable for both types. The postfix notation expression stacks are recognised as a platform and programming language independent method to describe, store in computer memory and evaluate general systems of differential and algebraic equations of any size. Each mathematical operation and its operands are described by a specially designed data structure, and every equation is transformed into an array of these structures (a Compute Stack). Compute Stacks are evaluated by a stack machine using a Last In First Out queue. The stack machine is implemented in the DAE Tools modelling software in the C99 language using two Application Programming Interface (APIs)/frameworks for parallelism. The Open Multi-Processing (OpenMP) API is used for parallelisation on general purpose processors, and the Open Computing Language (OpenCL) framework is used for parallelisation on streaming processors and heterogeneous systems. The performance of the sequential Compute Stack approach is compared to the direct C++ implementation and to the previous approach that uses evaluation trees. The new approach is 45% slower than the C++ implementation and more than five times faster than the previous one. The OpenMP and OpenCL implementations are tested on three medium-scale models using a multi-core CPU, a discrete GPU, an integrated GPU and heterogeneous computing setups. Execution times are compared and analysed and the advantages of the OpenCL implementation running on a discrete GPU and heterogeneous systems are discussed. It is found that the evaluation of model equations using the parallel OpenCL implementation running on a discrete GPU is up to twelve times faster than the sequential version while the overall simulation speed-up gained is more than three times.

Introduction

Most of engineering problems can be described by a system of non-linear (partial-) differential and algebraic equations. One of the methods to solve this type of problems is by using the Equation-Based approach (Barton & Pantelides, 1993). According to this approach, all equations and variables which constitute the model representing the process are generated and gathered together. Then, equations are solved simultaneously using a suitable mathematical algorithm (Morton, 2003). In the equation-based approach equations are given in an implicit form as functions of state variables and their derivatives, degrees of freedom (the system variables that may vary independently), and parameters. A typical Equation-Based simulation, optimisation, parameter estimation and sensitivity analysis software requires a numerical integration of a system of differential-algebraic equations (DAE) and integration of sensitivity equations (if local gradient based sensitivities are requested) by a suitable DAE solver. DAE solvers perform the following computationally intensive tasks (Fig. 1): (1) numerical integration in time (requires evaluation of equations residuals), (2) linear algebra operations (mostly BLAS L1 vector operations and some of BLAS L2 matrix-vector operations), (3) solution of systems of linear equations (direct solvers require evaluation of a Jacobian matrix while the iterative ones need a preconditioner), (4) integration of sensitivity equations (local gradient-based sensitivity analysis methods such as the Forward and Adjoint methods require evaluation of sensitivity residuals). Parallelisation of these tasks lead to a faster numerical solution.

Figure 1 Computationally intensive tasks performed by DAE solvers.

A large number of state of the art parallel linear algebra libraries are freely available such as AMD CPU Libraries (Advanced Micro Devices, Inc.; Santa Clara, CA, United States.) and Intel Math Kernel Library (Intel Corporation; Santa Clara, CA, United States) and several parallel linear solvers such as SuperLU_MT (Li, 2005), PARDISO (Schenk, Wächter & Hagemann, 2007) and Intel Pardiso and Iterative Sparse Solvers (Intel Corporation; Santa Clara, CA, United States). Thus, the focus in this is work is on parallel evaluation of residuals, Jacobian matrix and sensitivity residuals on shared memory computing platforms: (a) general purpose processors (traditional multi-core CPUs and manycore devices i.e. Xeon Phi), (b) streaming processors (General Purpose Graphics Processing Units, GPGPU and Field Programmable Gate Arrays, FPGA) and (c) heterogeneous computing systems (i.e. CPU + GPU and CPU + FPGA).

In general, model equations can be specified using the following approaches (again, only the problems that can be described by a system of partial or ordinary differential and algebraic equations are of interest): Using general-purpose programming languages such as C/C++ or Fortran to implement user-supplied functions for one of available suites for scientific applications such as SUNDIALS (Hindmarsh et al., 2005), Trilinos (Heroux et al., 2005) and PETSC (Balay et al., 2015).

Using modelling languages such as Ascend (Piela et al., 1991), gPROMS (Barton & Pantelides, 1994), Modelica (Fritzson & Engelson, 1998) and Dymola (Elmqvist, 1978).

Using multi-paradigm numerical languages such as Matlab (The MathWorks, Inc.; Natick, MA, United States), Mathematica (Wolfram Research, Inc.; Champaign, IL, United States) and Maple (Waterloo Maple, Inc.; Waterloo, Ontario, Canada).

Using higher-level fourth-generation languages (Python) and the modelling concepts provided by a software such as DAE Tools (Nikolić, 2016).

Using Computer Aided Engineering (CAE) software for finite element analysis and computational fluid dynamics such as COMSOL Multiphysics (COMSOL, Inc.; Stockholm, Sweden), ANSYS Fluent/CFX (Ansys, Inc.; Canonsburg, PA, United States), Abaqus (Dassault Systemes; Vélizy-Villacoublay, France), deal.II (Bangerth, Hartmann & Kanschat, 2007), libMesh (Kirk et al., 2006) and OpenFOAM (The OpenFOAM Foundation; London, United Kingdom).

The compiled user-supplied functions often produce the fastest computation. However, the process is error prone and it is difficult to specify expressions for analytical derivatives required for Jacobian/preconditioner matrix and sensitivity residuals. The source code of modelling languages is typically parsed into an Abstract Syntax Tree (AST). The produced AST can be transformed into a simulator-specific data structure or used to generate C source code as in OpenModelica (Fritzson et al., 2005) and JModelica (Akesson et al., 2010). Other modelling software such as DAE Tools (Nikolić, 2016) use the operator overloading technique to produce a tree-like data structure called an Evaluation Tree (ET). CAE software perform a discretisation of partial differential equations on a specified grid. On unstructured grids, the results of discretisation using the Finite Element (FE) or Finite Volume (FV) methods are the mass and stiffness matrices and load vectors. On structured grids, the results of discretisation using the Finite Difference (FD) method are the stencil data (nodes arrangement and their coefficients). Then, matrix-vector/matrix-matrix operations or stencil codes are used to obtain the results.

Evaluation of user-supplied functions, automatically generated source code and simulator-specific data structures can be easily performed on general purpose processors, either sequentially or in parallel. The parallel evaluation is fairly straightforward: every process/thread operates on its part of the array of equations. However, evaluation on streaming processors is rather difficult. Stream computing differs from traditional computing in that the system processes a sequential stream of elements: a kernel is executed on each element of the input stream and the result stored in an output stream. Hence, applications must be specially designed to effectively take advantage of the technology. For instance, the tree-like data structures are represented by non-contiguous blocks of memory and their evaluation typically requires recursive function calls which are not supported on all types of streaming processors. On the other hand, kernels for matrix-vector and matrix-matrix operations as well as stencil codes are widely available for virtually all Application Programming Interface (API) and hardware platforms. However, the general non-linear DAE systems cannot be represented in this way.

The main idea in this work is to implement a method to transform a general set of non-linear differential-algebraic equations into a stream of data that can be processed in parallel on both general purpose and streaming processors. In contrast to problems that can be described by a single system of finite element/finite volume/finite difference equations the focus in this work is on DAE systems that might include mixed/multiple coupled FE/FV/FD equations with additional ordinary differential and algebraic equations. Such mixed non-linear systems of equations are often found in multi-scale models and typically cannot be represented using a single mass/stiffness matrix and load vector nor evaluated using the stencil kernels.

The proposed approach is to transform equation expressions from infix notation into a Reverse Polish Notation (postfix notation) expression stack. For instance, the expression in infix notation x + y can be represented in postfix notation as xy + and stored in computer memory as an array of three items: [x, y, binary operator + ]. Expression stacks can be evaluated by a stack machine using a Last In First Out (LIFO) queue and in general are available for all platforms (due to their simplicity). Therefore, the postfix notation expression stacks are found as an efficient method to describe, store in computer memory and evaluate general DAE systems on all computing devices. Each mathematical operation and its operands are described by a specially designed data structure. This structure is very compact (to reduce memory transfers) and contains the op code and other data required for evaluation within the DAE Tools context. Every equation is represented by an array of these structures (called a Compute Stack in DAE Tools terminology). Since it is just a stream of binary data it can be read and used from any programming language. Evaluation of equations using the postfix notation expression stacks is obviously slower than compiled C code. However, the memory latency can be hidden in streaming processors by employing a large number of light-weight ‘threads’ with fast context switching (as shown in the Results section).

Implementation of a Compute Stack Machine requires an API and a runtime library to work on target platforms. There is a large number of available software API, libraries and languages that support shared memory multiprocessing and listing them all is out of scope of this work. Open Multi-Processing (OpenMP, http://www.openmp.org) has been chosen as one of the most widely used API for general purpose processors on shared memory systems. Open Computing Language (OpenCL, https://www.khronos.org/opencl) has been selected for streaming processors since it supports all hardware platforms of interest (CPU, GPGPU and FPGA) while its main competitor NVidia CUDA is available only on NVidia hardware. Regarding the programming language, OpenMP supports C/C++ and Fortran while OpenCL supports a static subset of C99 and a static subset of C++14 (since version 2.1). Therefore, C99 is the most suitable language for implementation as it is supported by both frameworks and all versions of the standards.

An overview of DAE Tools simulation utilising compute stack machines on different computing hardware is shown in Fig. 2. At the moment, simulations are executed on the host computer and the integration in time and solution of linear systems are performed on the CPU. The additional computing devices are used as accelerators for evaluation of model equations. A modelling software generates an array of Compute Stacks (one for every equation). All required data arrays are sent to a Compute Stack Evaluator (a class providing a common interface for evaluation of residuals, Jacobian and sensitivity residuals). The Compute Stack Evaluator implementation launches multiple OpenMP threads or OpenCL kernels, each of them running a Compute Stack Machine on the selected hardware.

Figure 2 DAE Tools simulation using a Compute Stack Evaluator and compute kernels.

An implementation of a Compute Stack Machine in C99 using OpenCL framework provides the following functionality/benefits: (a) parallel evaluation of any system of equations of any size, (b) support for all important computing platforms (including heterogeneous systems), (c) superior performance of streaming processors compared to the evaluation on general purpose processors, (d) straightforward scaling up to any number of equations and (e) straightforward switching to a different computing platform.

The article is organised in the following way. First, the required data structures and the OpenMP and OpenCL implementations in C99 are presented. Then, three medium scale models are benchmarked using the following computation setups: (1) sequential and OpenMP runs on multi-core CPU using the ET approach, (2) sequential and OpenMP runs on multi-core CPU using the Compute Stack approach, (3) OpenCL runs on multi-core CPU, discrete NVidia GPU and integrated Intel GPU using the Compute Stack approach, (4) OpenCL runs on heterogeneous multi-core CPU/discrete GPU system using the Compute Stack approach and (5) OpenCL runs on heterogeneous multi-core CPU/discrete GPU/integrated GPU system using the Compute Stack approach. The integration times are compared and analysed. Finally, a summary of the most important capabilities and benefits of the proposed implementation and directions for future work are given in the last section.

Implementation

Data structures

In DAE Tools, the operator overloading technique is used to transform the model equations given in implicit (acausal) form into a form suitable for evaluation. The standard mathematical operations and functions are re-defined to operate on a modified ADOL-C (Walther & Griewank, 2012) class adouble, which has been extended to contain simulator-specific information. In the existing approach, equations are transformed into a tree-like data structure called an Evaluation Tree. Evaluation Trees consist of a hierarchy of nodes where each node represents either an operand data or a mathematical operation. All node classes are derived from the base adNode abstract C++ class and in a simplified form given in the source code Listing 1. adNode class defines an interface for operations on ETs such as evaluation, units consistency check and export into the Latex or MathML format. Operand data nodes contain information about constants, parameters, variables and their time derivatives (adConstantNode, adRuntimeVariableNode and adRuntimeTimeDerivativeNode classes). Mathematical operation nodes represent unary (+, −) and binary operators (+, −, *, /, **) and standard mathematical unary and binary functions available in <math.h>: sqrt, pow, log, log10, exp, min, max, floor, ceil, abs, sin, cos, tan, asin, acos, atan, sinh, cosh, tanh, asinh, acosh, atanh, atan2 and erf (adUnaryNode and adUnaryNode classes). Unary nodes contain an operand node and a unary function to perform. Binary nodes contain two operand nodes (left and right) and a binary function to perform. During operations on an ET the node tree structure is traversed by calling functions on the top level node and on operand nodes in a recursive fashion.

Listing 1 ET related C++ classes (simplified).

/* Base abstract class */ class adNode { public:   virtual ∼adNode(){}   virtual adouble_t Evaluate(const daeExecutionContext* EC) const = 0;   virtual quantity GetQuantity() const = 0;   virtual string  SaveAsLatex(const daeNodeSaveAsContext* c) const = 0; }; /* Constants */ class adConstantNode : public adNode { public:   quantity value; }; /* Variable values */ class adRuntimeVariableNode : public adNode { public:   size_t overallIndex;   size_t blockIndex; }; /* Variable time derivatives */ class adRuntimeTimeDerivativeNode : public adNode { public:   size_t order;   size_t overallIndex;   size_t blockIndex; }; /* Unary operators and functions: −, sqrt, log, log10, exp, sin, cos, ... */ class adUnaryNode : public adNode { public:   adNode*       operand;   daeeUnaryFunctions function; }; /* Binary operators and functions: +, −, *, /, **, pow, atan2, min, max */ class adBinaryNode : public adNode { public:   adNode*        leftOperand;   adNode*        rightOperand;   daeeBinaryFunctions  function; };

However, recursive function calls are not supported by the OpenCL framework and a non-contiguous layout of nodes in memory is not compatible with the concept of OpenCL memory buffers. Therefore, the model equations need to be transformed into a suitable form. In this work, the expression stacks in Reverse Polish Notation (postfix notation) are proposed as a platform and programming language independent method to describe, store in computer memory and evaluate general systems of differential and algebraic equations of any size. An expression stack data structure in DAE Tools is called a Compute Stack. It represents an array of adComputeStackItem_t objects given in the source code Listing 2. Transformation of ETs is performed in a similar way as their evaluation. The Compute Stack arrays are automatically generated in DAE Tools software by traversing down the hierarchy of nodes in the ET data structure and adding the adComputeStackItem_t objects to an array in a predefined order for all operations and their operands.

Listing 2 Compute Stack related C99 data structures.

/* op codes */ typedef enum {   eOP_Unknown = 0,   eOP_Constant,   eOP_Time,   eOP_InverseTimeStep,   eOP_Variable,   eOP_DegreeOfFreedom,   eOP_TimeDerivative,   eOP_Unary,   eOP_Binary }daeeOpCode; /* real_t is defined as a single or a double precision floating point type. */ #define real_t double typedef struct adComputeStackItem_ {   uint8_t opCode;     /* Operation to perform */   uint8_t function;    /* Unary or binary function code */   uint8_t resultLocation; /* lvalue, value or rvalue LIFO stacks */   uint32_t size;      /* Size of the compute stack array */   union data_   {     real_t value;     /* For constants */     struct dof_indexes_  /* For degrees of freedom */     {       uint32_t overallIndex;       uint32_t dofIndex;     }dof_indexes;     struct indexes_   /* For variables (algebraic and differential) */     {       uint32_t overallIndex;       uint32_t blockIndex;     }indexes;   }data; }adComputeStackItem_t; /* Auxiliary data structures */ typedef struct adouble_ {   real_t value;   real_t derivative; } adouble_t; typedef struct daeComputeStackEvaluationContext_ {   real_t  currentTime;   real_t  inverseTimeStep;   uint32_t equationCalculationMode;   uint32_t sensitivityParameterIndex;   uint32_t jacobianIndex;   uint32_t numberOfVariables;   uint32_t startEquationIndex;   uint32_t numberOfEquations;   uint32_t startJacobianIndex;   uint32_t numberOfJacobianItems;   uint32_t numberOfDOFs;   uint32_t numberOfComputeStackItems;   uint32_t valuesStackSize;   uint32_t lvaluesStackSize;   uint32_t rvaluesStackSize; }daeComputeStackEvaluationContext_t; typedef struct adJacobianMatrixItem_ {   uint32_t equationIndex;   uint32_t overallIndex;   uint32_t blockIndex; } adJacobianMatrixItem_t;

The adComputeStackItem_t structure is very compact: its size is only 16 bytes (when aligned) and contains the following information: opCode (1 byte) is an integer defining an operation to be performed; function (1 byte) is an integer defining the mathematical operator/function for unary and binary items; resultLocation (1 byte) is an integer defining a location where to store the results (lvalue, rvalue or value LIFO stack); size (4 bytes) is an integer defining the size of the Compute Stack and it is always stored in the first item of the array; data (8 bytes) is a union holding a float value for constants, or indexes structure for variable values and time derivatives, or dof_indexes structure for values of degrees of freedom. As an example, the equation specified in implicit form: (1) x[0]−x[1]x[2]−x[3]+1.2⋅sin(x[0])=0

can be represented as an ET in Fig. 3 and as a Compute Stack in Fig. 4 (where oi represents the variable overallIndex and bi its blockIndex used to identify variables in data arrays).

Figure 3 The Evaluation Tree data structure representing Eq. (1).

Figure 4 The Compute Stack data structure representing Eq. (1).

Evaluation of a single equation

Evaluation of a single ET is performed in a recursive fashion starting with the top level node and traversing down the tree by processing its operand nodes.

A single Compute Stack is evaluated by a Compute Stack Machine using a LIFO queue. As an input, the Compute Stack Machine that calculates an equation residual (depicted in Fig. 5) requires the daeComputeStackEvaluationContext_t object, a Compute Stack array as a stream of contiguous data and random access data arrays with variable values (x), time derivatives (dx/dt) and degrees of freedom (y). daeComputeStackEvaluationContext_t structure contains all run-time information required by a Compute Stack Machine such as the current time, number of variables and sizes of input arrays. Evaluation of a Jacobian item requires an additional input array of adJacobianMatrixItem_t structures, while evaluation of a sensitivity residual requires two additional input arrays: sensitivity values and sensitivity derivatives.

Figure 5 The Compute Stack Machine.

An implementation of a Compute Stack Machine is fairly simple since it performs only a limited number of operations. It uses a LIFO queue composed of adouble_t objects. The adouble_t structure is used for auto-differentiation and holds a value and a derivative data. The Compute Stack Machine is implemented as a function evaluateComputeStack in C99. The algorithm (in a simplified form for clarity) is given in the source code Listing 3 while the auxiliary data structures in the source code Listing 2.

Listing 3 Compute Stack Machine implementation in C99 (a simplified algorithm).

adouble_t evaluateComputeStack(const adComputeStackItem_t*      computeStack,                     daeComputeStack EvaluationContext_t EC,                     const real_t*               dofs,                     const real_t*               values,                     const real_t*               timeDerivatives,                     const real_t*               svalues,                     const real_t*               sdvalues) {   for(int i = 0; i < computeStackSize; i++)   {     /* Take the item from the array. */     const adComputeStackItem_t item = computeStack[i];     /* Initialise a temporary adouble_t object to zero. */     adouble_t result = {0.0, 0.0};     if(item.opCode == eOP_Constant)     {      /* result.value is set to item.data.value      * result.derivative is zero. */     }     else if(item.opCode == eOP_Time)     {      /* result.value is set to EC.currentTime      * result.derivative is zero. */     }     else if(item.opCode == eOP_InverseTimeStep)     {      /* result.value is set to EC.inverseTimeStep      * result.derivative is zero. */     }     else if(item.opCode == eOP_Variable)     {      /* result.value is a variable value and set to values[blockIndex]      * result.derivative is set to:      * − 1.0 if overallIndex == EC.jacobianIndex      * − svalues[blockIndex] if overallIndex == EC.sensitivityParameterIndex      * − otherwise zero */     }     else if(item.opCode == eOP_TimeDerivative)     {      /* result.value is a variable derivative and set to timeDerivatives[blockIndex]      * result.derivative is set to:      * − EC.inverseTimeStep if overallIndex == EC.jacobianIndex      * − sdvalues[blockIndex] if overallIndex == EC.sensitivityParameterIndex      * − otherwise zero */     }     else if(item.opCode == eOP_DegreeOfFreedom)     {      /* result.value is a degree of freedom time and set to dofs[dofIndex]      * result.derivative is set to:      * − 1.0 if overallIndex == EC.sensitivityParameterIndex      * − otherwise zero */     }     else if(item.opCode == eOP_Unary)     {      /* The operand value is popped from the stack,      * unary function performed and the result pushed onto the stack. */     }     else if(item.opCode == eOP_Binary)     {      /* The left and right operands are popped from the stack,      * binary function performed and the result pushed onto the stack. */     }     /* Push the result onto the stack. */  }  /* Pop the final result from the stack and return. */ }

Evaluation of systems of equations

Systems of equations defined using the ET approach are stored in memory as an array of smart pointers to adNode-derived class objects. Parallel evaluation of systems of equations using the ET approach is rather straightforward: a single OpenMP parallel for loop is used to launch a team of threads, where each thread processes its chunk of equations. The C++ implementation is given in the source code Listing 4.

Listing 4 Evaluation of residuals using the ET approach and OpenMP API.

/* A system of equations is stored as an array: std::vector<adNodePtr> evaluationTrees. */ /* Set number of threads (or leave the default value). */ omp_set_num_threads(num_threads); /* Start the parallel OpenMP for loop. */ #pragma omp parallel for firstprivate(EC) for(int ei = 0; ei < numberOfEquations; ei++) {   adNode* node = evaluationTrees[ei];   /* The flag EC.equationCalculationMode defines the operation to perform:   * evaluation of residuals, Jacobian matrix or sensitivity residuals.   * Here, we calculate residuals. */   adouble_t residual = node−>Evaluate(&EC);   residuals[ei] = residual.value; }

Systems of equations defined using the Compute Stack approach are stored in memory as a single one-dimensional array of adComputeStackItem_t objects populated with Compute Stacks from all equations. The reason is OpenCL memory model where memory buffers can only hold contiguous blocks of generic data. Therefore, multidimensional arrays must be represented as one-dimensional ones. Since the size of individual Compute Stacks is not constant an additional array of integers are required that mark a starting index for each equation’s Compute Stack. In addition, the information required to evaluate a sparse Jacobian matrix are stored as an array of adJacobianMatrixItem_t objects. The adJacobianMatrixItem_t structure is given in the source code Listing 2 and contains the equation index and the variable block and overall indexes which identify a row and a column of the item in the Jacobian matrix and the variable with respect to which a derivative is requested.

Parallel evaluation of systems of equations using the Compute Stack approach is performed through a common interface called a Compute Stack Evaluator. The interface is specified in the adComputeStackEvaluator_t abstract C++ class given in the source code Listing 5. Compute Stack Evaluator implementations utilising a particular framework and/or targeting different computing devices are developed by deriving a new class from the base class. adComputeStackEvaluator_t class interface is designed with a support for heterogeneous computing in mind. Basically, it can contain nested evaluators targeting different computing hardware, each evaluator processing the specified chunk of equations. Two different Compute Stack Evaluator implementations have been developed: (a) using OpenMP API and (b) using OpenCL framework. OpenCL implementation can use a single or multiple OpenCL-enabled devices for heterogeneous computing. All implementations use the Compute Stack Machine function presented in the source code Listing 3. OpenMP implementation is given in the source code Listing 6 and OpenCL kernel in the source code Listing 7.

Listing 5 Compute Stack Evaluator abstract C++ class.

class adComputeStackEvaluator_t { public:  virtual ∼adComputeStackEvaluator_t(){}  /* Compute Stack related arrays are copied to the compute device   * only once (during the initialisation phase). */  virtual void Initialize(bool              calculateSensitivities,                 size_t           numberOfVariables,                 size_t           numberOfEquationsToProcess,                 size_t           numberOfDOFs,                 size_t           numberOfComputeStackItems,                 size_t           numberOfJacobianItems,                 size_t           numberOfJacobianItemsToProcess,                 adComputeStackItem_t*  computeStacks,                 uint32_t*         activeEquationSetIndexes,                 adJacobianMatrixItem_t* computeStackJacobianItems) = 0;  /* The arrays required for evaluation must be copied to the device   * before every computation. */  virtual void EvaluateResiduals(daeComputeStackEvaluationContext_t EC,                     real_t*                 dofs,                     real_t*                 values,                     real_t*                 timeDerivatives,                     real_t*                 residuals) = 0;  virtual void EvaluateJacobian(daeComputeStackEvaluationContext_t EC,                    real_t*                 dofs,                    real_t*                 values,                    real_t*                 timeDerivatives,                    real_t*                 jacobianItems) = 0;  virtual void EvaluateSensitivityResiduals(daeComputeStackEvaluationContext_t EC,                           real_t*                 dofs,                           real_t*                 values,                           real_t*                 timeDerivatives,                           real_t*                 svalues,                           real_t*                 sdvalues,                           real_t*                 sresiduals) = 0; };

Listing 6 Evaluation of residuals using the Compute Stack approach and OpenMP API.

void EvaluateResiduals(const adComputeStackItem_t*     computeStacks,                const uint32_t*             activeEquationSetIndexes,                daeComputeStackEvaluationContext_t EC,                const real_t*              dofs,                const real_t*              values,                const real_t*              timeDerivatives,                const real_t*              residuals) {  /* Systems of equations are represented as a single array populated with  * Compute Stacks from all equations:  * adComputeStackItem_t computeStacks[numberOfComputeStackItems].  * Starting indexes for each equation’s Compute Stack are stored in:  * uint32_t activeEquationSetIndexes[numberOfEquations]. */  /* Set number of threads (or leave the default value). */  omp_set_num_threads(num_threads);  /* Start the parallel OpenMP for loop. */  #pragma omp parallel for firstprivate(EC)  for(int ei = 0; ei < numberOfEquations; ei++)  {    /* Locate the current equation stack in the array of all compute stacks. */    uint32_t firstIndex              = activeEquationSetIndexes[ei];    const adComputeStackItem_t* computeStack = &computeStacks[firstIndex];    /* Evaluate the compute stack. */    adouble_t residual = evaluateComputeStack(computeStack,                             EC,                             dofs,                             values,                             timeDerivatives,                             NULL,                             NULL);    /* Set the value in the residuals array. */    residuals[ei] = residual.value;  } }

Listing 7 OpenCL kernel for evaluation of a single residual using the Compute Stack approach.

void ___kernel EvaluateResidual(___global adComputeStackItem_t*   computeStacks,                    ___global uint32_t*          activeEquationSetIndexes,                    daeComputeStackEvaluationContext_t EC,                    ___global real_t*           dofs,                    ___global real_t*           values,                    ___global real_t*           timeDerivatives,                    ___global real_t*           residuals) {   /* Get the global thread id. */   int residualIndex = get_global_id(0);   /* Get the equation index to process by this thread. */   int equationIndex = EC.startEquationIndex + get_global_id(0);   /* Locate the current equation stack in the array of all compute stacks. */   uint32_t firstIndex               = activeEquationSetIndexes[equationIndex];   ___global adComputeStackItem_t* computeStack = &computeStacks[firstIndex];   /* Evaluate the compute stack. */   ___global real_t* null_data = 0;   adouble_t residual = evaluateComputeStack(computeStack,                            EC,                            dofs,                            values,                            timeDerivatives,                            null_data,                            null_data);   /* Set the value in the residuals array. */   residuals[residualIndex] = residual.value; }

In the OpenMP implementation, every thread evaluates a chunk of the total number of equations, one at the time. In the OpenCL implementation, every thread evaluates only a single equation. Each thread executes a for loop where mathematical operations are performed in a single IF block controlled by the op_code (as given in the source code Listing 3). On GPUs, the thread divergence should not ever occur since the control flow does not depend on the thread ID and all threads execute the very same block of code (which does contain conditional jumps). At the moment, it is left to OpenCL compilers to optimise the IF block for a particular platform.

The OpenCL implementation of the Compute Stack Evaluator interface requires memory transfers between the CPU and the device: (1) the Compute Stack array, the active equation set indexes and the Jacobian information data are copied to a device only once during the initialisation (16 Ncs + 4 Neq + 12 Nnz bytes), (2) variable values and derivatives are copied before every evaluation (16 Neq bytes), (3) residuals and Jacobian values are copied from the device after every evaluation (8 Neq bytes for residuals and 8 Nnz bytes for Jacobian evaluation). Here, Neq is the number of equations/variables in the DAE system, Ncs is the size of the ComputeStack array (for all equations), and Nnz is the number of non-zero items in the Jacobian matrix. The current OpenCL implementation always explicitly copies the data between CPU-owned and device-owned buffers although in some instances where a computing device shares memory with the CPU (i.e. integrated GPU) it is possible to use a zero-copy buffer to avoid movement of data between different locations in the same memory.

Benchmarks

The performance of ET and Compute Stack approaches were evaluated by benchmarking the numerical solutions of three medium-scale models. The list and the description of all computing setups are given in Table 1. Cases 3 and 4 use the old ET approach and all other use the Compute Stack approach. Cases 2 and 4 utilise the OpenMP API while Cases 5–12 utilise the OpenCL framework. The heterogeneous cases used different proportions of the work done by individual OpenCL devices: (Case 8) 30% on the CPU and 70% on the NVidia GPU, (Case 9) 50% on the CPU and 50% on the NVidia GPU, (Case 10) 70% on the CPU and 30% on the NVidia GPU, (Case 11) 40% on the CPU, 40% on the NVidia GPU and 20% on the Intel HD GPU, and (Case 12) 30% on the CPU, 60% on the NVidia GPU and 10% on the Intel HD GPU. The hardware configuration consists of Intel i7-6700HQ CPU (four cores/eight threads at 2.6 GHz, 8 GB of RAM, 34.32 GFLOPS peak double precision, 128-bit bus width and 34.1 GB/s max memory bandwidth), an integrated Intel HD 530 GPU (24 execution units at 350 MHz, 256 MB of RAM, 33.6 GFLOPS peak double precision), and a discrete NVidia GeForce GTX 950M GPU (640 execution units at 914 MHz, 2 GB of RAM, 36.56 GFLOPS peak double precision, 128-bit bus width and 32 GB/s max memory bandwidth). The number of equations (Neq), the number of non-zero items in the Jacobian matrix (the total number Nnz=∑i=1NeqNnz[i] and the average number per equation Nnz/equation), the number of Compute Stack items (the total number Ncs=∑i=1NeqNcs[i] and the average number per equation Ncs/equation) and the average number of Compute Stack items for evaluation of a single row of the Jacobian matrix (Ncs/jacob_row=1Neq∑i=1NeqNnz[i]Ncs[i]) for all benchmarks are given in Table 2.

Table 1 Computational setups.

Case	Name	Description	
1	CPU Seq	Sequential on Intel CPU	
2	CPU OpenMP	OpenMP on Intel CPU	
3	CPU Seq (ET)	Sequential using the Evaluation Tree approach on Intel CPU	
4	CPU OpenMP (ET)	OpenMP using the Evaluation Tree approach on Intel CPU	
5	CPU OpenCL	OpenCL on Intel CPU	
6	GPU (HD)	OpenCL on Intel HD GPU	
7	GPU	OpenCL on NVidia GPU	
8	CPU + GPU (1)	OpenCL on Intel CPU (30%) and NVidia GPU (70%)	
9	CPU + GPU (2)	OpenCL on Intel CPU (50%) and NVidia GPU (50%)	
10	CPU + GPU (3)	OpenCL on Intel CPU (70%) and NVidia GPU (30%)	
11	CPU + GPU × 2 (1)	OpenCL on Intel CPU (40%), NVidia GPU (40%) and Intel GPU (20%)	
12	CPU + GPU × 2 (2)	OpenCL on Intel CPU (30%), NVidia GPU (60%) and Intel GPU (10%)	

Table 2 Size of the benchmarked models.

	Neq	Nnz	Ncs	Nnz/equation	Ncs/equation	Ncs/jacob_row	
Benchmark 1	46,948	205,218	9,199,459	4.37	196	1,475	
Benchmark 2	20,000	355,216	15,554,304	17.76	778	13,902	
Benchmark 3	36,864	952,278	7,122,536	25.83	193	5,318	

All benchmark models were implemented in Python using the DAE Tools v1.8.0 software (Nikolić, 2016). DAE Tools is free software released under the GNU General Public Licence. The source code, the installation packages and more information about the software can be found on the website (http://www.daetools.com). For integration of DAE systems in time the software uses the variable-step variable-order backward differentiation formula available in SUNDIALS IDAS DAE solver (Hindmarsh et al., 2005). Systems of linear equations are solved using SuperLU and SuperLU_MT solvers (Li, 2005). The benchmarks are carried out in 64-bit Debian Stretch GNU/Linux and compiled using the gcc 6.3 compiler, OpenMP 4.5 from the GOMP library, OpenCL 1.2 from NVidia CUDA 9.0 with v384.90 display driver, and OpenCL 2.0 from Intel GPU/CPU driver package for Linux r5.0.

Benchmark 1: transient two-dimensional Burgers equations, structured grid

The model describes a viscous fluid flow presented in Appendix C of Salari & Knupp (2000) and defined by the Burgers equations given in Eq. (2). The Burgers equations are spatially discretised using the centered FD Method.

(2) dudt+d(uu)dx+d(uv)dy−ν(d2udx2+d2udy2)=Su(x,y)dvdt+d(vu)dx+d(vv)dy−ν(d2vdx2+d2vdy2)=Sv(x,y)

Here, u and v are two components of the velocity field, ν is kinematic viscosity, and Su and Sv are the source terms.

This problem uses a formal code verification technique: the Method of Manufactured Solutions (Salari & Knupp, 2000) for checking the correctness of the numerical software. The method tests the software capabilities in full generality and therefore results in a high degree of confidence that all coding mistakes which prevent the equations being solved correctly have been identified. The procedure is the following: (1) a function representing the solution is selected (the manufactured solution), (2) the new source term for the original problem is computed and (3) the original problem is solved with the new source term. The computed numerical solution should be equal to the manufactured one. In this work, the quality of the solutions are checked using the most rigorous acceptance criteria available: the normalised global errors and the order of accuracy.

The manufactured solution in this example is given by: (3) um(x,y)=u0(sin(x2+y2+w0t)+ε)vm(x,y)=v0(cos(x2+y2+w0t)+ε)

and the new source terms are: (4) Su(x,y)=dumdt+d(umum)dx+d(umvm)dy−ν(d2umdx2+d2umdy2)Sv(x,y)=dvmdt+d(vmum)dx+d(vmvm)dy−ν(d2vmdx2+d2vmdy2)

Equations are distributed on a two-dimensional rectangular grid (−0.1, 0.7) × (0.2, 0.8) with 120 × 96 grid points. Input parameters are: u0=1.0ms, v0=1.0ms, w0 = 0.1, ν=0.7m2s, ɛ = 0.001. The Dirichlet boundary conditions on all four edges are set using the manufactured solution. The initial conditions are set using the manufactured solution with w0 = 0.0. The system is integrated for 90 s and the outputs are taken every 2 s. The source code is given in the Supplemental Source Code Listing 1 (benchmark_1.py file) and on DAE Tools website (http://www.daetools.com/docs/tutorials-cv.html#tutorial-cv-4).

The normalised global errors (Eu for the horizontal and Ev for the vertical velocity component) and orders of accuracy (pu for the horizontal and pv for the vertical velocity component) for the steady-state solution and different mesh sizes (grids used: 10 × 8, 20 × 16, 40 × 32 and 80 × 64) are given in Table 3. The results show a decrease in the normalised global error with the decrease in mesh size and the order of accuracy approaching the theoretical order 2, as expected.

Table 3 The normalised global errors and the orders of accuracy for the steady-state solution.

Mesh	10 × 8	20 × 16	40 × 32	80 × 64	
||Eu||	1.92e-04	4.80e-05	1.20e-05	3.00e-06	
||Ev||	4.80e-04	1.20e-04	3.00e-05	7.50e-06	
pu		1.996	1.999	2.000	
pv		1.994	1.999	2.000	

In addition, for benchmarking purposes, the same model is re-implemented in plain C++. The functions for evaluation of residuals and the Jacobian matrix use the same auto-differentiation method as in the other approaches: operator overloading technique adopted from ADOL-C. The time for a sequential evaluation of 1,000 residuals and Jacobian matrix in a single loop is measured and compared for both the C++ and the Compute Stack implementations. This way the effect of other parts of the simulation code is minimised and only evaluation of model equations is benchmarked. The source code is given in the Supplemental Source Code Listing 1 (cxx-cs-benchmark directory). The compilation instructions are located in the read_me.txt file. The software loads the Compute Stack and Jacobian data from the input data files, calls the C++ and ComputeStack evaluation functions in a loop for the given number of iterations and records the execution times.

Benchmark 2: transient two-dimensional Cahn–Hilliard equations, unstructured grid

The model describes the process of phase separation, where two components of a binary mixture separate and form domains pure in each component. The problem is defined by the Cahn–Hilliard equations given in Eq. (5).

(5) dcdt=D∇2μ μ   =c3−c−γ ∇2c

Here, D is the diffusion coefficient, c is the concentration, μ is the chemical potential and γ defines the length of the transition regions between the domains. The mesh is a simple square (0,100) × (0,100) with 100 × 100 elements. Input parameters are D = 1 and γ = 1. Boundary conditions for both c an μ are insulated boundary conditions (no flux on boundaries). Initial conditions are set to c(0) = 0.5 + cnoise where the noise cnoise is specified using the normal distribution with standard deviation of 0.1. The system is integrated for 500 s and the outputs are taken every 5 s. The source code is given in the Supplemental Source Code Listing 2 (benchmark_2.py file) and on DAE Tools website (http://www.daetools.com/docs/tutorials-fe.html#tutorial-dealii-3).

The Cahn–Hilliard equations are spatially discretised using the Finite Elements Method. In DAE Tools, deal.II (http://dealii.org) library is utilised for low-level tasks such as mesh loading, management of finite element spaces, degrees of freedom, assembly of the system stiffness and mass matrices and the system load vector, and setting the boundary conditions. The assembled system matrices are used to generate a set of equations in the following form: (6) [M]{x˙}+[A]{x}−{F}=0

where x and x˙ are vectors of state variables and their derivatives, M and A are mass and stiffness matrices and F is the load vector. The generated set of equations (in general case a DAE system) are solved together with the rest of equations in the model.

Benchmark 3: transient two-dimensional Stokes equations, unstructured grid

The model describes a fluid flow caused by the differences in buoyancy as a result of temperature differences (Stokes flow). Parts of the fluid that are hotter have a lower density and rise up while parts that are cooler are denser and sink down with gravity. The differences to the original problem are that the grid is not adaptive and no stabilisation method is used. The problem is defined by the Stokes equations given in Eq. (7). The Stokes equations are spatially discretised using the Finite Elements Method. (7)  −∇⋅(2ηε(u))+∇p=−ρβTg   −∇⋅u=0    dTdt+u⋅∇T−∇⋅k∇T=γ    

Here, u is the velocity field, p is the pressure, ɛ is the symmetric gradient of the velocity and T is temperature. The mesh is a simple square (0, 1) × (0, 1) with 96 × 96 elements. Input parameters are κ = 2 · 10−5, β = 10.0, η = 1.0 and ρ = 1.0. The source term in the heat balance equation (γ) is defined as three point-like heat sources near the bottom of the grid with the coordinates (0.3, 0.1), (0.45, 0.1) and (0.75, 0.1), the radius r = 1/32 and the value 1.0. Boundary conditions are no-slip no-flux conditions for the velocity, no-flux conditions for the pressure, and an insulated boundary condition for the temperature. Initial conditions are set to 0.0 for the temperature and 0.0 for the velocity. The system is integrated for 50 s and the outputs are taken every 0.5 s. The source code is given in the Supplemental Source Code Listing 3 (benchmark_3.py file) and on DAE Tools website (http://www.daetools.com/docs/tutorials-fe.html#tutorial-dealii-7). The problem is originally defined in the step-31 deal.II tutorial (https://www.dealii.org/8.5.0/doxygen/deal.II/step_31.html).

Results

The comparison of the performance of the Compute Stack, the ET and the direct C++ implementations of the Benchmark 1 model are given in Table 4. The time required to copy the inputs (variable values and derivatives) to OpenCL buffers and results from OpenCL buffers, given as percentage of the total time for evaluation of residuals, is presented in Table 5.

Table 4 Performance comparison of the ComputeStack, the Evaluation Tree and the C++ implementations of Benchmark 1 model.

	EvaluateResiduals	EvaluateJacobian	
C++	CS	ET	C++	CS	ET	
Average time/call, ms	48.10	69.70	364.30	353.30	507.57	1676.56	
Relative to C++	1.00	1.45	7.57	1.00	1.44	4.75	
Relative to CS	0.69	1.00	5.23	0.69	1.00	3.30	

Table 5 Time required for the data transfer to/from OpenCL buffers, given as percentage of the total time for evaluation of residuals.

	Case 5	Case 6	Case 7	
Benchmark 1:	Copy inputs	1.23	3.39	1.99	
	Copy outputs	0.49	2.52	1.19	
	Total	1.72	5.91	3.18	
Benchmark 2:	Copy inputs	0.45	1.10	0.61	
	Copy outputs	0.18	0.86	0.35	
	Total	0.63	1.96	0.96	
Benchmark 3:	Copy inputs	2.30	2.44	2.04	
	Copy outputs	0.66	1.69	1.27	
	Total	2.96	4.13	3.31	

Execution times for different phases of the numerical solution given as percents of the total integration time for the base case, Case 1 (CPU Seq) are given in Fig. 6. Four phases have been analysed: (1) time for evaluation of residuals, (2) time spent in a linear solver (excluding evaluation of a Jacobian matrix), (3) time for evaluation of a Jacobian matrix and (4) time spent in a DAE solver. Clearly, the first two benchmarks are dominated by the execution time of evaluation functions (approximately 85% of the total time). On the other hand, Benchmark 3 is dominated by the time spent in the linear solver, mostly for the sparse matrix factorisation (71% of the total time). This becomes more understandable if the numbers of non-zero items in the Jacobian matrix in Table 2 are compared: the number of items per equation in Benchmark 1 is approximately 4 while in Benchmark 3 it is 26. Therefore, both effects, evaluation of equations and solution of linear systems, need to be considered in Benchmark 3.

Figure 6 Case 1–Percentage of the integration time for individual phases of the numerical solution in Benchmark 1 (A), Benchmark 2 (B) and Benchmark 3 (C).

The maximum theoretical speed-ups for evaluation of model equations (compared to the sequential Case 1) can be estimated using the maximum peak performance for individual platforms. For instance, in Case 7 the theoretical speed-up is 36.56 GFlops/(34.32 GFlops/8 cores) = 8.52, while for Cases 2, 4 and 5 it is equal to the number of cores (eight in this work). The maximum theoretical overall simulation speed-ups (as a result of parallelisation of only model equations) compared to the sequential Case 1 can be calculated from the Amdahl’s law using the data from Fig. 6 and the maximum peak performance for individual platforms: 1/(1−p+p/s), where p is the portion of the solution that can be parallelised (in this case the sum of the time spent for evaluation of residuals and the Jacobian) and s is the maximum theoretical speed-up for evaluation of model equations. The maximum theoretical overall and speed-ups for evaluation of model equations are summarised in the Table 6.

Table 6 The maximum theoretical speed-ups for evaluation of model equations and the maximum overall simulation speed-ups (relative to the sequential Case 1).

	Maximum overall simulation speed-up	Maximum speed-up for model equations	
Case	Benchmark 1	Benchmark 2	Benchmark 3	
2, 4, 5	3.71	4.01	1.33	8.00	
6	3.68	3.98	1.32	7.83	
7	3.80	4.12	1.33	8.52	
8, 9, 10	4.64	5.16	1.36	16.52	
11, 12	5.02	5.64	1.37	24.35	

The speed-ups achieved in the Compute Stack approach compared to the old ET one, measured during the simulations, are summarised in Table 7. The maximum overall speed-ups observed in Case 7 (NVidia GPU) compared to Cases 1 (CPU Seq) and 4 (CPU OpenMP (ET)) are given in Table 8. Execution times (in seconds) and speed-ups of individual phases for the single-threaded linear solver (SuperLU) for Benchmark 1 are presented in Tables 9 and 10, for Benchmark 2 in Tables 11 and 12, and for Benchmark 3 in Tables 13 and 14. The speed-ups presented are given relative to the base case, Case 1 (sequential run on the CPU). In addition, all runs were repeated using the multi-threaded SuperLU_MT linear solver using OpenMP API. The time spent in the linear solver and the overall speed-ups (compared to the base case where the serial linear solver is used) for Benchmarks 1, 2 and 3 are presented in Tables 15–17, respectively.

Table 7 Speed-up observed in the ComputeStack compared to the Evaluation Tree approach.

	EvaluateResiduals	EvaluateJacobian	
Benchmark	1	2	3	1	2	3	
CPU Seq vs. CPU Seq (ET)	6.25	4.55	7.69	4.00	2.63	4.00	
CPU OpenMP vs. CPU OpenMP (ET)	4.29	3.54	5.74	4.24	2.41	2.99	

Table 8 Maximal overall speed-up observed in Case 7 (NVidia GPU) relative to Cases 1 (CPU Seq) and 4 (CPU OpenMP (ET)).

Benchmark	1	2	3	
GPU vs. CPU Seq	2.84	3.16	1.26	
GPU vs. CPU OpenMP (ET)	4.66	3.95	1.75	

Table 9 Benchmark 1–execution times in seconds for individual phases of the numerical solution.

Case	Run	EvaluateResiduals	EvaluateJacobian	Linear solver	DAE solver	
1	CPU Seq	19.57 (57.1%)	9.06 (26.4%)	5.39 (15.7%)	0.28 (0.8%)	
2	CPU OpenMP	8.94 (48.2%)	2.67 (14.4%)	6.64 (35.8%)	0.31 (1.7%)	
3	CPU Seq (ET)	121.51 (74.4%)	35.91 (22.0%)	5.52 (3.4%)	0.32 (0.2%)	
4	CPU OpenMP (ET)	38.53 (68.6%)	11.28 (20.1%)	5.99 (10.7%)	0.33 (0.6%)	
5	CPU OpenCL	8.64 (46.3%)	4.34 (23.2%)	5.40 (28.9%)	0.29 (1.6%)	
6	GPU (HD)	14.64 (47.6%)	5.63 (18.3%)	9.64 (31.3%)	0.87 (2.8%)	
7	GPU	4.68 (38.8%)	1.98 (16.5%)	5.11 (42.4%)	0.29 (2.4%)	
8	CPU + GPU (1)	7.03 (41.5%)	2.68 (15.8%)	6.94 (41.0%)	0.29 (1.7%)	
9	CPU + GPU (2)	9.32 (52.0%)	2.91 (16.2%)	5.40 (30.1%)	0.30 (1.7%)	
10	CPU + GPU (3)	9.37 (49.6%)	3.80 (20.1%)	5.41 (28.7%)	0.30 (1.6%)	
11	CPU + GPU × 2 (1)	8.45 (47.2%)	2.35 (13.1%)	6.71 (37.5%)	0.39 (2.2%)	
12	CPU + GPU × 2 (2)	7.37 (43.4%)	2.11 (12.4%)	6.76 (39.8%)	0.75 (4.4%)	

Table 10 Benchmark 1–speed-up achieved in the evaluate functions and the overall speed-up.

Case	Run	EvaluateResiduals	EvaluateJacobian	Overall	
1	CPU Seq	1.00	1.00	1.00	
2	CPU OpenMP	2.19	3.39	1.85	
3	CPU Seq (ET)	0.16	0.25	0.21	
4	CPU OpenMP (ET)	0.51	0.80	0.61	
5	CPU OpenCL	2.27	2.09	1.84	
6	GPU (HD)	1.34	1.61	1.11	
7	GPU	4.18	4.56	2.84	
8	CPU + GPU (1)	2.79	3.38	2.02	
9	CPU + GPU (2)	2.10	3.11	1.91	
10	CPU + GPU (3)	2.09	2.38	1.82	
11	CPU + GPU × 2 (1)	2.32	3.86	1.92	
12	CPU + GPU × 2 (2)	2.65	4.29	2.02	

Table 11 Benchmark 2–execution times in seconds for individual phases of the numerical solution.

Case	Run	EvaluateResiduals	EvaluateJacobian	Linear solver	DAE solver	
1	CPU Seq	84.20 (50.3%)	59.37 (35.5%)	23.43 (14.0%)	0.38 (0.2%)	
2	CPU OpenMP	34.44 (42.1%)	19.42 (23.7%)	27.57 (33.7%)	0.44 (0.5%)	
3	CPU Seq (ET)	377.63 (67.3%)	157.98 (28.2%)	24.70 (4.4%)	0.43 (0.1%)	
4	CPU OpenMP (ET)	122.70 (61.8%)	46.90 (23.6%)	28.32 (14.3%)	0.47 (0.2%)	
5	CPU OpenCL	40.63 (42.8%)	29.00 (30.5%)	24.94 (26.3%)	0.41 (0.4%)	
6	GPU (HD)	74.52 (46.4%)	39.43 (24.5%)	45.76 (28.5%)	1.02 (0.6%)	
7	GPU	24.71 (46.6%)	5.01 (9.5%)	22.90 (43.2%)	0.38 (0.7%)	
8	CPU + GPU (1)	25.35 (42.9%)	7.83 (13.2%)	25.06 (42.4%)	0.86 (1.5%)	
9	CPU + GPU (2)	25.59 (41.3%)	10.38 (16.8%)	25.54 (41.2%)	0.44 (0.7%)	
10	CPU + GPU (3)	24.42 (37.2%)	15.33 (23.3%)	25.49 (38.8%)	0.46 (0.7%)	
11	CPU + GPU × 2 (1)	33.33 (44.2%)	16.17 (21.4%)	25.53 (33.8%)	0.45 (0.6%)	
12	CPU + GPU × 2 (2)	23.52 (40.1%)	9.27 (15.8%)	25.30 (43.2%)	0.50 (0.8%)	

Table 12 Benchmark 2–speed-up achieved in the evaluate functions and the overall speed-up.

Case	Run	EvaluateResiduals	EvaluateJacobian	Overall	
1	CPU Seq	1.00	1.00	1.00	
2	CPU OpenMP	2.44	3.06	2.04	
3	CPU Seq (ET)	0.22	0.38	0.30	
4	CPU OpenMP (ET)	0.69	1.27	0.84	
5	CPU OpenCL	2.07	2.05	1.76	
6	GPU (HD	1.13	1.51	1.04	
7	GPU	3.41	11.85	3.16	
8	CPU + GPU (1)	3.32	7.59	2.83	
9	CPU + GPU (2)	3.29	5.72	2.70	
10	CPU + GPU (3)	3.45	3.87	2.55	
11	CPU + GPU × 2 (1)	2.53	3.67	2.22	
12	CPU + GPU × 2 (2)	3.58	6.40	2.86	

Table 13 Benchmark 3–execution times in seconds for individual phases of the numerical solution.

Case	Run	EvaluateResiduals	EvaluateJacobian	Linear solver	DAE solver	
1	CPU Seq	16.33 (16.5%)	11.57 (11.7%)	70.65 (71.3%)	0.53 (0.5%)	
2	CPU OpenMP	8.12 (9.4%)	5.22 (6.0%)	72.43 (83.8%)	0.65 (0.7%)	
3	CPU Seq (ET)	121.76 (49.6%)	47.20 (19.2%)	76.05 (31.0%)	0.58 (0.2%)	
4	CPU OpenMP (ET)	46.11 (33.6%)	15.72 (11.5%)	74.77 (54.5%)	0.63 (0.5%)	
5	CPU OpenCL	8.46 (9.7%)	6.13 (7.0%)	72.31 (82.7%)	0.56 (0.6%)	
6	GPU (HD)	21.99 (19.0%)	11.59 (10.0%)	81.27 (70.3%)	0.68 (0.6%)	
7	GPU	6.57 (8.3%)	1.59 (2.0%)	70.11 (89.0%)	0.50 (0.6%)	
8	CPU + GPU (1)	7.27 (9.1%)	1.52 (1.9%)	70.24 (88.3%)	0.54 (0.7%)	
9	CPU + GPU (2)	6.53 (7.9%)	3.11 (3.8%)	72.71 (87.7%)	0.55 (0.7%)	
10	CPU + GPU (3)	7.17 (8.5%)	3.96 (4.7%)	72.76 (86.2%)	0.53 (0.6%)	
11	CPU + GPU × 2 (1)	21.24 (20.7%)	8.27 (8.1%)	71.57 (69.9%)	1.35 (1.3%)	
12	CPU + GPU × 2 (2)	18.91 (19.9%)	4.00 (4.2%)	71.20 (74.9%)	0.91 (1.0%)	

Table 14 Benchmark 3–speed-up achieved in the evaluate functions and the overall speed-up.

Case	Run	EvaluateResiduals	EvaluateJacobian	Overall	
1	CPU Seq	1.00	1.00	1.00	
2	CPU OpenMP	2.01	2.21	1.15	
3	CPU Seq (ET)	0.13	0.25	0.40	
4	CPU OpenMP (ET)	0.35	0.74	0.72	
5	CPU OpenCL	1.93	1.89	1.13	
6	GPU (HD)	0.74	1.00	0.86	
7	GPU	2.49	7.29	1.26	
8	CPU + GPU (1)	2.25	7.59	1.25	
9	CPU + GPU (2)	2.50	3.71	1.20	
10	CPU + GPU (3)	2.28	2.92	1.17	
11	CPU + GPU × 2 (1)	0.77	1.40	0.97	
12	CPU + GPU × 2 (2)	0.86	2.89	1.04	

Table 15 Benchmark 1–the time in seconds spent in the multi-threaded linear solver and the overall speed-up.

Case	Run	Linear solver (MT)	Overall speed-up	
1	CPU Seq	4.94 (14.4%)	1.01	
2	CPU OpenMP	5.53 (30.8%)	1.91	
3	CPU Seq (ET)	5.26 (3.2%)	0.21	
4	CPU OpenMP (ET)	5.55 (10.0%)	0.62	
5	CPU OpenCL	5.10 (26.2%)	1.76	
6	GPU (HD)	9.14 (30.3%)	1.14	
7	GPU	5.06 (42.1%)	2.86	
8	CPU + GPU (1)	5.17 (29.2%)	1.94	
9	CPU + GPU (2)	5.20 (28.1%)	1.85	
10	CPU + GPU (3)	5.44 (27.0%)	1.70	
11	CPU + GPU × 2 (1)	5.09 (30.5%)	2.06	
12	CPU + GPU × 2 (2)	5.26 (35.0%)	2.28	

Table 16 Benchmark 2–the time in seconds spent in the multi-threaded linear solver and the overall speed-up.

Case	Run	Linear solver (MT)	Overall speed-up	
1	CPU Seq	16.56 (10.4%)	1.05	
2	CPU OpenMP	18.97 (27.8%)	2.45	
3	CPU Seq (ET)	17.09 (3.4%)	0.33	
4	CPU OpenMP (ET)	20.64 (11.3%)	0.92	
5	CPU OpenCL	16.72 (19.1%)	1.91	
6	GPU (HD)	37.02 (24.5%)	1.11	
7	GPU	16.55 (35.4%)	3.58	
8	CPU + GPU (1)	17.18 (34.5%)	3.36	
9	CPU + GPU (2)	18.34 (38.6%)	3.53	
10	CPU + GPU (3)	17.70 (29.4%)	2.78	
11	CPU + GPU × 2 (1)	21.21 (29.6%)	2.34	
12	CPU + GPU × 2 (2)	17.93 (35.3%)	3.29	

Table 17 Benchmark 3–the time in seconds spent in the multi-threaded linear solver and the overall speed-up.

Case	Run	Linear solver (MT)	Overall speed-up	
1	CPU Seq	38.32 (57.3%)	1.48	
2	CPU OpenMP	42.65 (75.8%)	1.76	
3	CPU Seq (ET)	41.02 (21.1%)	0.51	
4	CPU OpenMP (ET)	41.81 (44.1%)	1.04	
5	CPU OpenCL	39.70 (72.2%)	1.80	
6	GPU (HD)	49.87 (60.4%)	1.20	
7	GPU	39.16 (81.9%)	2.07	
8	CPU + GPU (1)	37.90 (79.8%)	2.04	
9	CPU + GPU (2)	39.96 (79.3%)	1.97	
10	CPU + GPU (3)	40.11 (78.3%)	1.93	
11	CPU + GPU × 2 (1)	40.02 (60.9%)	1.51	
12	CPU + GPU × 2 (2)	40.26 (59.2%)	1.46	

Verification of the Compute Stack approach using the Method of Manufactured Solutions is performed in Benchmark 1 and both OpenMP and OpenCL implementations perform as expected. Additional code verification tests can be found on DAE Tools website in the Code Verification Tests section (http://www.daetools.com/docs/tutorials-cv.html).

Discussion

The performance of the Compute Stack approach was first compared to the direct C++ implementation of the Benchmark 1 model (Table 4). It was found that the Compute Stack approach is 45% slower for evaluation of residuals and 44% for evaluation of the Jacobian. It must be kept in mind that these results are approximate since neither the Compute Stack nor the C++ implementation are optimised for maximum performance. Although the Compute Stack approach can never achieve better performance than the direct C++ implementation, the Compute Stack Machine and Evaluator can be optimised for each platform. At the moment, the same code is used for all platforms.

The Compute Stack approach brings significant performance improvements compared to the ET approach. The speed-ups for both sequential and parallel cases using identical hardware and API are given in Table 7. Evaluation of residuals is 4.5–7.7 times faster for sequential runs and 3.5–5.7 times for OpenMP runs, while evaluation of Jacobian matrix is 2.6–4.0 times faster for sequential runs and 2.4–4.2 times for OpenMP runs (both using eight threads). It is difficult to attribute the causes for the performance improvement in a definitive way. However, the possible reasons could be the fact that ETs are represented by a large number of non-contiguous blocks of memory, occupy almost an order of magnitude more memory, and require millions of calls to virtual functions that must be resolved at runtime.

Regarding the effect of the memory transfer between the CPU and the computing devices, it can be seen from Table 5 that it does not significantly affects performance: it requires 0.6–3.0% of the time for evaluation of residuals for Intel CPUs, 2.0–5.9% for the integrated Intel HD GPU and 1.0–3.3% for the discrete NVidia GPU. The reason is that the most of the data are copied to a device only once during the initialisation and only the inputs (variable values and derivatives) and results (residuals or Jacobian) need to be transferred during every evaluation of equations. Furthermore, the time for memory transfer can in some cases be reduced by using the pinned/mapped memory buffers or even completely avoided if a computing device shares memory with the CPU (i.e. integrated GPU).

The general trend in all Benchmarks (Tables 10, 12 and 14) is that the best performance is always achieved in Case 7 where a single discrete NVidia GPU was used: the achieved speed-ups in Benchmarks 1–3 are 4.18, 3.41 and 2.49 for evaluation of residuals and 4.56, 11.85 and 7.29 for evaluation of the Jacobian. The maximum theoretical speed-up in Case 7 is 8.52 (Table 6). The speed-ups for evaluation of the Jacobian are always higher than speed-ups for evaluation of residuals since a much larger amount of computation is required and the hardware is better utilised. This is especially evident in Cases 7–9 where all or a larger portion of the calculations are done on a discrete GPU. The computation load for evaluation of residuals is proportional to the average number of Compute Stack items per equation (Ncs/equation in Table 2): 196, 778 and 193 for Benchmarks 1–3, respectively. The computation load for evaluation of the Jacobian is proportional to the average number of Compute Stack items for evaluation of a single row in the Jacobian matrix (Ncs/jacob_row in Table 2): 1,475, 13,902 and 5,318 for Benchmarks 1–3, respectively. Consequently, the hardware resources per single evaluation are significantly better utilised. In addition, the speed-up for evaluation of the Jacobian in Benchmark 2 (11.85) is higher than the theoretical one (8.52). Again, it is difficult to specify the exact cause. However, the probable reason is the memory latency that has a greater negative effect on the CPU evaluation. The evaluation procedure in the Compute Stack Machine requires a for loop where data must often be fetched from the global memory before a mathematical operation can be performed. When a variable value/derivative is required for calculation, the CPU must wait for the data since the memory location based on the variable index is not known in advance. The memory latency can be hidden on the GPU by using a large number of work-items (‘threads’): while work-items in one work-group wait for the data, the scheduler can switch to another work-group to perform mathematical operations. This is particularly true for Benchmark 2 where, based on a high Ncs/jacob_row value, a large amount of data must be fetched from the global memory causing a very poor performance in the sequential case. On the other hand, the speed-ups achieved on the Intel multi-core CPU in Case 2 (OpenCL implementation) are 2.19, 2.44 and 2.01 for evaluation of residuals and 3.39, 3.06 and 2.21 for evaluation of the Jacobian, and in Case 5 (OpenMP implementation) are 2.27, 2.07 and 1.93 for evaluation of residuals and 2.09, 2.05 and 1.89 for evaluation of the Jacobian. The maximum theoretical speed-up in these cases is eight (Table 6). Case 7 is between 24% and 287% faster compared to Case 2 (95% in average) although NVidia GPU can deliver only 7% more FLOPS. Therefore, the Compute Stack approach appears to be more suitable for GPUs than multi-core CPUs. However, it must be kept in mind that the computation performance also largely depends on the memory performance (since variable values must be fetched from the global GPU memory or from the main CPU memory) and the complexity of model equations (i.e. Benchmark 2 contains a large number of pow function evaluations and the performance of mathematical functions may vary between platforms). In addition, in order to reach the maximum performance the Compute Stack Machine and the Evaluator must be further optimised.

Regarding the overall simulation speed-up, the best performance is again observed in Case 7 (Table 8): (a) the maximum overall speed-ups compared to the sequential Case 1 are 2.84, 3.16 and 1.26, and (b) the maximum overall speed-ups compared to Case 4 are 4.66, 3.95 and 1.75 for Benchmarks 1, 2 and 3, respectively. Since the maximum overall simulation speed-ups are 3.80, 4.12 and 1.33 for Benchmarks 1, 2 and 3 (Table 6), the achieved performance is between 5.2% and 25.3% lower than theoretical. The structure of model equations plays a very important role for determination of the maximum overall speed-up. For instance, as a result of the fact that only 28.2% of the integration time is spent on evaluation of model equations, the maximum overall speed-up in Benchmark 3 is only 33% on the hardware used in this work, and 71.3% of the code is still run sequentially.

Overall, the performance is generally in the following order: Case 7 (GPU) > Cases 8–10 (CPU + GPU) > Cases 11–12 (CPU + GPU × 2) > Case 2 (CPU OpenMP) > Case 5 (CPU OpenCL). In the models used in this work, the heterogeneous setups are always somewhat slower than a single discrete GPU. In addition, heterogeneous setups that perform a larger portion of work on a discrete NVidia GPU (Cases 8 and 12) always perform better. Parallel Compute Stack evaluation using OpenMP is typically faster than using OpenCL on the identical hardware: overall, OpenMP simulations perform up to 20% faster. The performance of all heterogeneous configurations are far from the maximum theoretical. For instance, the speed-ups in the fastest heterogeneous Case 8 are 2.79, 3.32 and 2.25 for evaluation of residuals and 3.38, 7.59 and 7.59 for evaluation of the Jacobian while the maximum theoretical speed-up is 16.52. Again, it is difficult to attribute the definite causes for the observed trends. The possible reason is that the current implementation of the multi-device Compute Stack Evaluator is not optimised for best performance and more tests are required to make a definite conclusion. In addition, the time required for management of multiple threads, each running a Compute Stack Evaluator, can contribute to the higher execution times than expected.

The Stokes equations in Benchmark 3 create a complex system of coupled differential equations. This results in a large number of non-zero items in the Jacobian matrix and affects the matrix factorisation in the linear solver. The consequence is that the numerical solution is dominated by the time spent in the linear solver and the overall speed-ups in parallel cases is fairly modest: up to 26% in Case 7 (NVidia GPU). Therefore, additional runs have been performed using the multi-threaded linear solver (SuperLU_MT) with eight threads. As it can be seen from Table 17, this leads to much better performance: the maximum overall combined speed-ups is now 2.07 (again in Case 7). As for the first two benchmarks, use of a multi-threaded linear solver produce only a modest improvement since the solution process is dominated by the evaluation functions: less than 10% percents in Benchmark 1 and up to 40% in Benchmark 2.

Conclusions

In this work, the methodology for parallel evaluation of general systems of differential and algebraic equations on shared memory computing devices such as general purpose processors (traditional multi-core CPUs and manycore devices), streaming processors (GPGPU and FPGA) and heterogeneous systems have been presented.

The postfix notation expression stacks (Compute Stacks) have been proposed as a platform and programming language independent method to describe, store in computer memory and evaluate general DAE systems of any size. Evaluation of Compute Stacks is performed by Compute Stack Machines implemented in C99 language. Parallel evaluation of systems of equations in DAE Tools modelling software is performed using the Compute Stack Evaluator interface. Two implementations have been developed using: (a) the OpenMP API for general purpose processors, and (b) the OpenCL framework for streaming processors and heterogeneous systems. Both implementations have been verified using the Method of Manufactured Solutions.

The performance of the Compute Stack approach has been evaluated by benchmarking three medium-scale models: (1) a viscous fluid flow model described by the transient Burgers equations, (2) a phase separation model described by the transient Cahn–Hilliard equations and (3) a fluid flow model described by the transient Stokes equations. A total of 12 different configurations utilising a multi-core CPU, an integrated Intel GPU, a discrete NVidia GPU and heterogeneous computing setups have been analysed.

Four phases of the numerical solution have been analysed: (1) time for evaluation of residuals, (2) time spent in a linear solver, (3) time for evaluation of a Jacobian matrix and (4) time spent in the DAE solver. Execution times of individual phases of the numerical solution and the overall simulation speed-ups have been compared and discussed. Typically, more than 95% of the total integration time is spent on evaluation of equations and the solution of linear systems. The solution process in some examples is dominated by evaluation of equations while in others by matrix factorisation in the linear solver.

The maximum theoretical speed-ups for evaluation of model equations and the maximum theoretical overall simulation speed-ups have been estimated and compared to the benchmark results in this work. Evaluation of model equations using the sequential Compute Stack approach has been compared to the direct C++ and the old ET implementations. It has been found that the Compute Stack approach is 45% slower than the direct C++ implementation and more than five times faster than the old one. Regarding the parallel implementations, the highest performance has been achieved using the OpenCL framework running on a discrete NVidia GPU: evaluation of model equations is up to 12 times faster than the sequential version while the maximum overall simulation speed-up is more than three times. It has been discovered that the Compute Stack approach performs significantly faster on the GPU than on the multi-core CPU with the similar peak power (95% in average). In addition, it has been observed that the overall simulation performance is generally in the following order: single discrete GPU > heterogeneous CPU+GPU > heterogeneous CPU+GPUx2 > multi-core CPU using OpenMP > multi-core CPU using OpenCL.

Future work will be focused on optimisation of the Compute Stack Machine performance and implementations for additional types of computing devices (such as Xeon Phi and FPGA).

Supplemental Information

Supplemental Information 1 Python and C++ source code for the Benchmark 1.

Click here for additional data file.

Supplemental Information 2 Python source code for the Benchmark 2.

Click here for additional data file.

Supplemental Information 3 Python source code for the Benchmark 3.

Click here for additional data file.

Additional Information and Declarations

Competing Interests

Author Contributions

Data Deposition

The authors declare that they have no competing interests.

Dragan D. Nikolić analysed the data, prepared figures and/or tables, performed the computation work, authored or reviewed drafts of the paper, approved the final draft.

The following information was supplied regarding data availability:

http://www.daetools.com,

http://sourceforge.net/projects/daetools.

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
