# Peer review of "Parallelisation of equation-based simulation programs on heterogeneous computing systems"

_PeerJ Computer Science, doi:10.7717/peerj-cs.160_

## Round 0.1 · original submission · Major Revisions

In your revision, please address all referee comments.

Reviewer 1 ·

Basic reporting

Overall, the quality of presentation and use of language was high. The manuscript is well-written and well-structured. The motivation for the work was clearly explained. The diagrams are nicely made and used to good effect in explaining the work. There was no trouble following the flow of the presentation.

While I did not read the manuscript carefully with the intent of identifying all mistakes in grammar, I found some minor uses of language that I suggest could be improved in the following way:

In the abstract on lines 24 and 25 - the joining of OpenMP API and OpenCL framework to utilised is somewhat grammatically incorrect. May I suggest breaking the sentence up into two sentences. Perhaps something along the lines of "...*the* DAE Tools modeling software in *the* C99 language using two APIs/frameworks for parallelism: *The* OpenMP API is used for parallelization on general purpose processors, and the OpenCL framework is used for parallelization on streaming processors and heterogeneous systems."

30 - "is discussed" instead of just "discussed"

109 - "equation expressions" instead of "equations expressions" (no s on equation)

118 - "the" before "DAE Tools context"

123 - "as shown in the Results section" instead of "as it has been shown in the Results section"

141 - comma after OpenCL kernels

165 - comma before which

576 - 583 - "a" in front of every "multi-core CPU"

576 - I believe it should read Cases 1 and 2 instead of Cases 1 and 3, judging from Table 1

578 - Likewise Cases 3 and 4 instead of Cases 2 and 4

584,etc. - replace "on" with "at" before every "clock frequency"

605 - This experiment is discretized using finite differences, so perhaps "grid points" instead of "elements"

647 - Colon replaced by comma, as well as a comma after "linear systems"

649, 652, etc. - all cases of "speed-up" should be "speed-ups"

673 - comma before "each requiring"

673 - "resolution" instead of "the resolution"

674 - comma before while

690 - "outweigh" instead of "out-weight"

717 - the in front of each item, i.e. "the OpenMP API", "the OpenCL framework"

721 - a/an in front of each item, e.g. "a viscious fluid model"

733 - "the solution of linear systems", i.e. add "the"

Experimental design

1:

The primary thing that I think needs some revision in this manuscript is the description of the experiments. First of all, not all of the details are given in the description. For example, all of the equations are time-dependent, but the time interval over which the solutions is evolved does not seem to be given.

Furthermore, the nature of DAE solver and linear solver are not described in the Benchmarks section. In the Results section, it eventually is stated that the linear solver is SuperLU, but I did not see any similar statement about the DAE solver. This means that the spatial discretization is described (finite-difference or finite-element), but the nature of the time stepping is not known. Is BDF? Runge-Kutta? Is a particular package being used? If so, which solver inside the package is chosen?

Furthermore, it is not clear which parts of the system are running on the GPU vs. the CPU. Clearly at least the equation evaluation is being done on the GPU, but what else? From the discussion in the Results section and Discussion section, one can infer that it is probably the case that the linear solver and DAE solver are running on the CPU. In other words, the problem is primarily run on the CPU and the evaluation of the equations is done using the GPU as an "accelerator". This is in contrast to a common approach in HPC where the problem remains on the GPU for the length of the problem without fine-grain movement of data between the CPU and GPU between components (e.g. between the DAE solver and linear solver). Each part of the system acts directly on the GPU data. It would be helpful to make that relationship clear as early as the Implementation section of the paper. Under what conditions is there transfer of data between the CPU and GPU?

In light of the fact that the equation evaluation is (probably) being done in an accelerator style, there is a question of whether a copy is being done with the integrated GPU (the Intel HD 530). Since that device shares memory with the CPU, it is possible to use a zero-copy buffer to avoid movement from a CPU-owned buffer to a GPU-owned buffer, which avoids a copy to different locations in the same memory. Clearly a copy is being made out of necessity with the GTX 950M, but is one being done with the HD 530?

The performance of streaming architectures is highly influenced by the bandwidth of memory. It would be helpful to list the memory bandwidth for each of the architectures described in the Benchmark section. In particular, the higher bandwidth of the dedicated GPU is probably the biggest reason for its superior performance over other cases. Furthermore, it would also be helpful to list the bus bandwidth between the CPU memory and the dedicated GPU memory, as this determines the cost of moving data between the CPU and GPU. Movement of data between memories is generally *the* highest overhead in heterogeneous systems.

In light of this, it would be helpful to list the final amount of data being moved back and forth between the CPU and GPU in each benchmark in terms of bytes (or some other direct measure of the data cost beyond requiring the reader to infer and calculate from equation unknowns etc.). The author has already made some effort in the Implementation section to describe the efficient use of memory in the design of the Compute Stack structure. Some discussion about how that adds up in the final system would be helpful for showing off the design.



2:

The performance of the Compute Stack approach is evaluated relative to the performance of the Evaluation Tree approach implemented in DAE Tools. In other words, the comparisons are being done between two implementations done by the same author in the same software suite. We see that the Compute Stack approach outcompetes the Evaluation Tree approach, so it's better than (the author's own) Evaluation Tree implementation, but how good is the Evaluation Tree implementation? From the manuscript alone, we have no way of knowing how good of a baseline the Evaluation Tree implementation is. To put it more simply, the performance results are relative improvements over an unknown (absolute) baseline. A reader sees that the Compute Stack GPU performance is considerably improved, but improved over something that was good or bad?

Of course this is being somewhat pedantic, and likely the implementation of the original Evaluation Tree is fine. The author says up front that direct use of C/C++ and Fortran are generally the fastest approach, and it's clear that some loss of performance for the sake of convenience is acceptable here. Furthermore, DAE Tools is being successfully used in applications, so it seems likely that the original Evaluation Tree implementation is not downright pathological.

Still, it would be very helpful to include something that grounds the performance in more absolute terms. If a reference can be made to another paper that compares performance with other software or that otherwise discusses the performance of the software, that would resolve the matter.

In my opinion, a particularly nice approach would be to go so far as to compare the performance to a direct C/C++ implementation. For the test problems used in this paper, a C/C++ or direct OpenCL implementation would be straight-forward, and it is likely that the author is using a package for the DAE solve (such as something like SUNDIALS) that can accept such direct implementations already. Of course, the direct implementation would probably be somewhat faster, but everyone expects and accepts that one is making a trade-off for convenience here. But it would clear up the issue that one is showing a speedup over something that is ultimately unknown, and it would also show that the cost of the trade-off for convenience is acceptable as a bonus.


3:

What versions of the compilers, APIs, and frameworks are being used?

Validity of the findings

In the Discussion section, there is some speculation about the underlying causes for the performance differences. For example, in the first paragraph the author points out that Evaluation Trees use a large number of non-contiguous blocks of memory that cannot fit into cache, that a large number of virtual function evaluations are required, and that there could be pipelining of instructions and hiding of memory latency in the Compute Stack case. Everything that the author states is plausible, but it is important to make very clear that all the suggestions are plausible but not proven. In general, performance analysis of codes is very tricky, and attributing causes in a definitive way is very difficult. I do not think a fine-grained attribution of causes is needed for demonstrating the value of this work in any case, so it is best to mostly avoid it. I do not think the Discussion section would be any less for having most of the speculation about causes simply removed. If it is kept, it should be stated explicitly that the attribution is hypothetical.

However, I take this opportunity to emphasize again that further detail about the movement of data between the CPU and GPU in the Implementation section would make the nature of the system more clear. The author highlights this particular cost in the Discussion section, noting appropriately that such a cost is a probable cause but not a proven one. However, this is the first place in the manuscript where this aspect of the system is enunciated and it is assumed that the reader understood this all along.

Additional comments

Reviewers were instructed that the paper should be evaluated strictly on scientific grounds and not on impact, but I do believe that with some modest refinement this paper will make a valuable contribution to the literature.

Reviewer 2 ·

Basic reporting

The English is clear and the explanation almost unambiguous; the state of the art is well presented. The goal and structure of the article are both good, although the Results section could be improved.

In particular, there is some confusion between the "types of computational setups that were benchmarked" (a list of 5 items) compared to the Table 1 named "computational setups", which has 12 items. I suggest to stick to the 12-items description, by extending the list in the text and by integrating the information about the proportions of work done. The same 12-items convention should be used in the Conclusions. Moreover, case #5 (i.e., CPU + OpenCL) is named "CPU" which is absolutely confusing, since it resembles case #1 (i.e., simple sequential CPU execution). A different naming is strongly suggested (e.g., CPU + OpenCL).

A few minors:
- Abstract: "It is found that the sequential Compute Stack approach is up to one order of magnitude faster than the old approach for evaluation of equations". It is not clear what "old" means in this context. The same comment is valid for the Discussion ("The Compute Stack approach brings significant performance improvements compared to the old one").
- Page 2/26: Procesing -> Processing
- Page 17/26: "u is the velocity field" <- symbol "u" is a vector and should be bold (see Equation 6).
- All execution time tables: please report the unit (seconds?)
- Bibliography: openmodelica -> OpenModelica. Also add: "46th Conference on Simulation and Modelling of the Scandinavian Simulation Society (SIMS2005), Trondheim, Norway"

Experimental design

The paper focuses on the comparison of DAE programs implemented using different computing architectures (i.e., CPU and GPU), including heterogeneous solutions (e.g., CPU+GPU). Moreover, the author compares two alternative formalisms for equations encoding, i.e., classic evaluation trees (ET) and computing stacks (CS). The research falls within the aims and scope of PeerJ Computer Science, and it is interesting for researchers involved in the development of high-performance scientific simulations.

Validity of the findings

According to the author, CSs are more adeguate than ETs on streaming processors since the evaluation of ETs "typically requires recursive function calls which are not supported on streaming processors". However, this sentence is true only in the case of OpenCL code, because Nvidia's CUDA supports recursive calls since the Fermi architecture. The author should clarify/correct this passage since CUDA could, in principle, execute ETs on the GPU.

CSs seem to be more efficient (as shown in Table 4) than ETs. Moreover, GPUs (even a modest integrated Intel) provide a decent speedup with respect to multithreaded code. However, the result with heterogeneous computation (i.e., GPU + CPU) is actually surprising, because it is supposed to be the fastest option since the CPU is generally idle during kernels' execution. The speed-up should be at least similar, but it is actually one half in the results shown in Table 7. This is an unexpected and interesting result that should be investigated better. The author states that "[t]he probable reason is that the time required to copy input arrays to multiple devices, start multiple OpenCL kernels, and copy the results back out-weight the higher theoretical computing capacity of multiple devices". However, the memory copy performed with simple GPU-based computation (i.e., case 7) should be more intensive, since *all* data is transferred back and forth over the (slow) PCI-ex bus. I suggest to investigate the issues of heterogeneous computation more carefully before making such a claim.

The speedup provided by GPU with respect to single-threaded sequential execution is puzzling as well. Even though it is true that the (theoretical) peak power is 33.6 and 36.56 GFlops for the CPU and GPU (respectively), the former can be reached only by using intenstive multi-threading. Thus, the 640 units of the GPU should provide a better speedup and instructions throughput than CPU-based single threaded execution. Maybe some optimization in the source code (e.g., by leveraging cached __constant memory for non-mutable data) could improve the results.

Reviewer 3 ·

Basic reporting

One of my biggest concerns is that the scope of "equation-based simulations" is simultaneously quite large and also quite vague. Do you mean "differential equations", or even more specifically "partial differential equations" or "ordinary differential equations"? The terminology used does not seem like a standard one to me, since all simulations are "equation-based" in some way. Similarly, the very first sentence seems like a big understatement—are there any engineering problems that aren't described using a system of non-linear (partial)-differential and/or algebraic equations?

Experimental design

The Benchmark 1 section references "Appendix C", but I cannot find any appendices.

I have a few questions about the method:
a) A single float "value" field is used in adComputeStackItem_t is used for constants; does this limit the complexity of operations? Or would operations with more than one constant just be broken into multiple structs?

b) To clarify, parallelizing the evaluation of an adComputeStackEvaluator_t object uses a single CPU thread to calculate a single equation? Does this also extend to the GPU parallelization, where a single GPU thread handles a single equation?

c) How are equations set up using the Compute Stack approach? Does something automatically generate the code shown, or is this done manually?

Regarding how the results are shown/discussed, I think it is a little confusing to choose the "CPU Seq" benchmark as the baseline for showing speedup etc.; instead, wouldn't one of the cases using the Evaluation Tree approach be a better baseline, since that is the prior method against which you are comparing? This issue also relates to how the methods are presented; in my first read, it was not 100% clear that the Compute Stack method is new and the Evaluation Tree approach is the existing approach against with CS is being compared. So, some more clarity on the objectives of the study would be helpful.

In the Benchmarks section, I think that each benchmarks is missing quite a bit of detail on how the governing equations are actually solved. In particular, benchmarks 2 and 3 evidently use the finite element method, but no additional detail is given. Furthermore, there is not enough discussion of the other components required to perform the benchmark studies, such as the linear solver (which appears in Figure 6 prior to its first mention).

(Minor comment: the description of the Method of Manufactured Solutions technique is a bit too strong; it is a good method for creating an analytical solution for verification purposes, but I don't think you can say it ensures "all coding mistakes which prevent the equations from being solved correctly have been identified"...)

Validity of the findings

While I don't doubt the validity of the findings, and think that the work is useful due to the improvement in performance shown, the manuscript does not quite explain why the method behaves as observed. For example, why do the performance trends shown happen?

Furthermore, how well does this method actually parallelize on the systems considered (thinking about, e.g., thread divergence on GPUs, vectorization efficiency on the SIMD devices)? The performance offered on the GPU benchmarks is not actually that high compared with the CPU-based ones, so I wonder how well this approach is actually suited for GPU execution. Some analysis and discussion of these topics would strengthen the manuscript from its current form, which mostly just shows the results of the method without much critical analysis. (There are some speculative comments in the Discussion section on page 23, but these aren't supported by any analysis.)

Additional comments

I appreciate the availability of the benchmark code listed in the Appendix, but for longevity purposes I suggest that it be archived using a more permanent service (e.g., Zenodo) and cited in the manuscript.

Furthermore, it isn't clear whether the DAE Tools software (which the research relies on) is openly available. At minimum it should be cited appropriately in the manuscript, including authors, version used, location where accessible; in the best case, the version used should be archived as described above for maximum reproducibility of the study.

Minor comment: the colors used for Figure 6 could be chosen a little better for accessibility (i.e., colorblindness).

---

## Round 0.2 · Minor Revisions

Thanks for your efforts in addressing the reviewer comments!
Please address the minor revision proposed by Reviewer 2.

Reviewer 1 ·

Basic reporting

As with the original submission, the overall presentation of this submission is excellent, the structure is well-organized and clear, and the use of English is of high quality.

I only make the following minor comment. English rules about whether to use "the" with nouns, particularly proper nouns that are capitalized, are somewhat murky. However, I believe the most common convention is to refer to cases, figures, and tables, etc. without "the" but using a capital letter, e.g. "Case 1" as opposed to "the Case 1". In particular, I believe the most common convention for "Case" follows the same rules that the manuscript already uses with tables, such as "in Table 2" on line 745.

Whether a particular usage could be considered incorrect is perhaps debatable, but certainly a convention should be applied consistently throughout a manuscript, at least as long as doing so does not hurt clarity or simplicity. It is a minor issue, but some lines use "the" with Case(s), such as lines 619 and 622, where as some do not, such as line 762. Furthermore, some lines do not capitalize "case", such as lines 751 and 754.

Some situations are tricky, such as "to the base Case 1" in line 767. The most technically correct usage would probably be "to the base case, Case 1,...", but that seems unnecessarily obtuse for the sake of rule following, and it seems fine to leave such cases as is.

Experimental design

The author has done a very thorough job of revising the experimental design and its presentation. All of my questions about the first submission have been well addressed. In particular, taking the extra effort to compare with plain C++ is much appreciated. In my opinion, the current state of the manuscript is quite satisfactory.

Validity of the findings

I sympathize that the author had to balance somewhat competing demands from the reviewers about attribution of causes. In my opinion, the current presentation is well balanced in that regard. Overall, all of my concerns with the first manuscript have been well addressed and there are no further ones.

Reviewer 2 ·

Basic reporting

The paper is now clear, self-contained, well written and fluid to read.

Experimental design

The paper is in scope with the journal. The research question is relevant for GPU-powered simulation based on ODEs.

Validity of the findings

The concerns expressed in the previous review have been addressed; however, the new analyses introduce new complexity and raised new doubts. Specifically, the author should provide a minimal explanation why, in Case #7, the actual speed-up achieved during the evaluation of residuals is 11.85x while the theoretical speed-up is just 8.52x. In general, performances higher than the theoretical peak processing power are unexpected and deserve additional analysis

Reviewer 3 ·

Basic reporting

no comment

Experimental design

no comment

Validity of the findings

no comment

Additional comments

Overall the author has satisfactorily addressed my original comments.

---

## Round 0.3 · accepted · Accept

I appreciate your careful and complete responses to the reviewer comments.